

# Analysis of structural relationship among the occupational dysfunction on the psychological problem in healthcare workers: a study using structural equation modeling

Mutsumi Teraoka[1,2] and Makoto Kyougoku[3]

[1] Graduate School of Health Sciences, Kibi International University, Japan
[2] Oosugi Hospital, Okayama, Japan
[3] Department of Occupational Therapy, School of Health Sciences, Kibi International University, Okayama, Japan

Corresponding author
Mutsumi Teraoka,
mutsu13t@gmail.com

## ABSTRACT

**Purpose.** The purpose of this study is to demonstrate the hypothetical model based on structural relationship with the occupational dysfunction on psychological problems (stress response, burnout syndrome, and depression) in healthcare workers.

**Method.** Three cross sectional studies were conducted to assess the following relations: (1) occupational dysfunction on stress response ($n = 468$), (2) occupational dysfunction on burnout syndrome ($n = 1,142$), and (3) occupational dysfunction on depression ($n = 687$). Personal characteristics were collected through a questionnaire (such as age, gender, and job category, opportunities for refreshment, time spent on leisure activities, and work relationships) as well as the Classification and Assessment of Occupational Dysfunction (CAOD). Furthermore, study 1 included the Stress Response Scale-18 (SRS-18), study 2 used the Japanese Burnout Scale (JBS), and study 3 employed the Center for Epidemiological Studies Depression Scale (CES-D). The Kolmogorov–Smirnov test, confirmatory factor analysis (CFA), exploratory factor analysis (EFA), and path analysis of structural equation modeling (SEM) analysis were used in all of the studies. EFA and CFA were used to measure structural validity of four assessments; CAOD, SRS-18, JBS, and CES-D. For examination of a potential covariate, we assessed the correlation of the total and factor score of CAOD and personal factors in all studies. Moreover, direct and indirect effects of occupational dysfunction on stress response (Study 1), burnout syndrome (Study 2), and depression (Study 3) were also analyzed.

**Results.** In study 1, CAOD had 16 items and 4 factors. In Study 2 and 3, CAOD had 16 items and 5 factors. SRS-18 had 18 items and 3 factors, JBS had 17 items and 3 factors, and CES-D had 20 items and 4 factors. All studies found that there were significant correlations between the CAOD total score and the personal factor that included opportunities for refreshment, time spent on leisure activities, and work relationships ($p < 0.01$). The hypothesis model results suggest that the classification of occupational dysfunction had good fit on the stress response (RMSEA = 0.061, CFI = 0.947, and TLI = 0.943), burnout syndrome (RMSEA = 0.076, CFI = 0.919,

and TLI = 0.913), and depression (RMSEA = 0.060, CFI = 0.922, TLI = 0.917). Moreover, the detected covariates include opportunities for refreshment, time spent on leisure activities, and work relationships on occupational dysfunction.

**Conclusion.** Our findings indicate that psychological problems are associated with occupational dysfunction in healthcare workers. Reduction of occupational dysfunction might be a strategy of better preventive occupational therapies for healthcare workers with psychological problems. However, longitudinal studies will be needed to determine a causal relationship.

## INTRODUCTION

Practitioners, educators, and researchers acknowledge occupational dysfunction as a major health problem in preventive occupational therapy (*Jackson et al., 1998*; *Mandel & Association AOT, 1999*; *Horowitz & Chang, 2004*; *Scaffa & Reitz, 2013*). Occupational dysfunction is defined as a negative experience emerging from an unsatisfactory lifestyle atmosphere; it includes occupational imbalance, occupational deprivation, occupational alienation, and occupational marginalization (*Teraoka & Kyougoku, 2014*; *Teraoka & Kyougoku, 2015*). Occupational marginalization is defined as impeding participation in daily activities (*Townsend & Wilcock, 2004*). Occupational deprivation is a lack of choices in daily activities that are beyond the individual's control (*Gail, 2000*). Occupational alienation is the failure to fulfill the inner needs in everyday activities (*Wendy, Christine & McKay, 2004*). Occupational imbalance is a loss of balance in engagement during daily activities (*Dana et al., 2010*).

Occupational dysfunction occurs not only among the disabled but also in healthy persons (*Kyougoku, 2010*; *Ishii, Kyougoku & Nagao, 2010*). It has been indicated that occupational dysfunction can occur in the absence of apparent medical disease (*Kyougoku, 2010*). According to a finding of an observational study of workers without obvious medical disease, 36% of workers have some occupational dysfunction (*Akiyama & Kyougoku, 2010*). Regarding occupational alienation, 43% of workers from the same study reported experiencing serious psychological problems (*Akiyama & Kyougoku, 2010*). Moreover, a report found that occupational dysfunction was observed in 75.4% of rehabilitation therapists without obvious medical disease, and occupational dysfunction showed a correlation with job stress (*Miyake et al., 2014*). People suffering from occupational dysfunction are unable to participate in day-to-day activities of work, leisure, self-care, and rest.

A previous study indicates that healthcare workers frequently experience occupational dysfunction and various psychological issues than other professionals, which includes stress response, burnout syndrome, and depression (*Akiyama & Kyougoku, 2010*; *Miyake et al., 2014*). Depression and burnout syndrome are due to an increase in stress response (*Irvine, 1997*; *Van Praag, 2004*). Job related stress response is defined as the harmful

physical and emotional responses that occur when the requirements of the job do not correlate with the capabilities, resources, or needs of the worker (*National Institute for Occupational Safety and Health, 1999*). Job related stress response is recognized as a major psychological issue for healthcare workers (*Cooper, Dewe & O'Driscoll, 2001*; *Wright, 2007*). Moreover, burnout syndrome is defined as a job related stress response that includes symptoms of exhaustion and indifference toward work (*Peterson et al., 2011*). Burnout syndrome influences the job related performance of the healthcare worker to collaborate with other team members under challenging circumstances (*Moghaddasi et al., 2013*). At the individual level, burnout syndrome is related to depression for the healthcare worker (*Drury et al., 2014*). Depression is defined as a problem with persistent feelings of sadness and emptiness and a loss of pleasure and interests (*Ishii, Kyougoku & Nagao, 2010*). In Japanese society, there is a recognized association between depressive mood and subsequent suicide among workers (*Tamakoshi et al., 2000*; *Takeuchi & Nakao, 2013*). Furthermore, over 60% of workers are reported to suffer from a stress related psychological problem in Japan (*Honda et al., 2014*). One of the primary causes of psychological problem among workers is attributed to difficult working conditions, such as heavy overtime work, understaffing, deadline pressure, relationship problems, and cost-cutting practices (*Kato et al., 2014*; *Denton et al., 2002*; *Schaefer & Moos, 1993*; *Seki & Yamazaki, 2006*). Many Japanese workers are classified as workaholics, which leads to fatigue and is also one of the causes of depression (*Seki & Yamazaki, 2006*; *Matsudaira et al., 2013*). There has been a growing concern about the psychological problems, especially among healthcare workers, because stress response, burnout syndrome, and depression are the most common work-related health problems in the healthcare profession (*National Institute for Occupational Safety and Health, 2008*).

However, there has been no previous study examining the impact of classification of occupational dysfunction on psychological problems, including stress response, burnout syndrome, and depression. A case study on occupational therapy has both suggested a causal association between occupational dysfunction and psychological problem (*Kyougoku, 2010*; *Ishii, Kyougoku & Nagao, 2010*). A theoretical study on occupational dysfunction has related to mental well-being (*Wilcock & Hocking, 2015*). Therefore, occupational dysfunction may predate the appearance of psychological problems in workers.

We hypothesize that occupational dysfunction has the impact of the structural relationship for the stress response, burnout syndrome, and depression of healthcare workers in hospitals (Fig. 1). The structural relationship is a snapshot of a point in time and observed findings suggest an influence on variables. In Study 1, we hypothesize that occupational dysfunction, as assessed by the Classification and Assessment of Occupational Dysfunction (CAOD), is associated with the Stress Response Scale-18 (SRS-18). In Study 2, we hypothesize that occupational dysfunction is associated with the Japanese Burnout Scale (JBS). In Study 3, we hypothesize that occupational dysfunction is associated with the Center for Epidemiological Studies Depression Scale (CES-D).

Moreover, we surmise that occupational dysfunction and psychological problems in healthcare workers is further influenced by personal factors, including age, gender, years

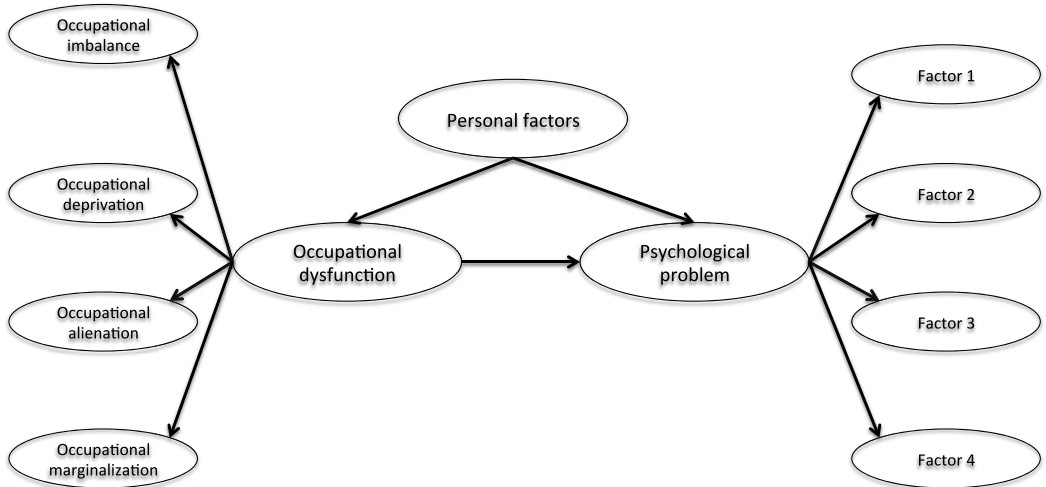

**Figure 1 The hypothesized model of structural relationship.** Note. Occupational dysfunction includes occupational imbalance, occupational deprivation, occupational alienation, and occupational marginalization. Psychological problems include stress response (Study 1), burnout syndrome (Study 2), and depression (Study 3). The purpose of three studies are to examine whether the hypothesis model can be reproduced. Personal factors include age, gender, years of work experience, job category, opportunities for refreshment, time spent on leisure activities, and work relationships.

of work experience, job category, opportunities for refreshment, time spent on leisure activities, and quality of work relationships.

In summary, this study demonstrated the hypothesis model of structural relationship that psychological problems are affected by occupational dysfunction among healthcare workers.

## Ethics statement

The Ethics Committee of Kibi International University and the research ethics committee of partnership hospitals approved all research protocol and informed consent procedures (No. 13–30). Written informed consent was obtained from all the participants. We provided participants with a letter explaining the outline and purpose of the study. Participants had the right to drop out of the research project at any time for any reason. We regarded the return of the survey sheet as consent for participation in this research. Survey sheets were returned in anonymous, sealed envelopes.

## Statistical software

SPSS Statistics (http://www.spss.com) were used for the sample characteristics and correlation analysis. Mplus 7.3 (http://www.statmodel.com) was used for the structural equation modeling (SEM) in all studies. The SEM is a comprehensive statistical analysis of the integration of path analysis and factor analysis (*Ullman & Bentler, 2003*). SEM aids the identification of structural or cause relationships (*Ullman & Bentler, 2003*). Mplus is a statistical modeling software for SEM, developed by *Muthen & Muthen (2012)* and is a flexible tool to analyze multivariate data.

## STUDY 1

### Purpose

The aim of this study is to test the hypothesis that job related stress response affects occupational dysfunction in healthcare workers (see Fig. 1). Moreover, this hypothesis model examines the effect of personal factors on job related stress response and occupational dysfunction.

### Methods

#### Participants

In total, there were 468 participants (21 doctors, 159 nurses, 52 physical therapists, 60 occupational therapists, and 176 other healthcare workers).

#### Measures

*Sample characteristics.* Demographic data were obtained from all participants. We assessed age, gender, job category, years of work experience, opportunities for refreshment, time spent on leisure activities, and work relationships.

*CAOD (Teraoka & Kyougoku, 2015).* We developed CAOD for measuring the classification of occupational dysfunction, based on the Occupation Based Practice 2.0 (OBP2.0) (Teraoka & Kyougoku, 2014; Teraoka & Kyougoku, 2015). Figure 2 demonstrates that OBP2.0 offers a conceptual foundation for the assessment and intervention in occupational dysfunction and belief conflict under various circumstances (Teraoka & Kyougoku, 2015). Using this model, the occupational therapy practitioner is able to enhance the occupational therapy effects in a person with occupational dysfunction (Teraoka & Kyougoku, 2015). Furthermore, the occupational therapy practitioner can help the client overcome belief conflicts by using the OBP2.0 (Teraoka & Kyougoku, 2014; Kyougoku, 2011). Thus, this model will be able to increase the quality of occupational therapy and teamwork. The CAOD measures occupational dysfunction in four domains: occupational marginalization (6 items), occupational imbalance (4 items), occupational alienation (3 items), and occupational deprivation (3 items). The CAOD comprises 16 items on a 7-point Likert scale (1 = strongly disagree, 7 = strongly agree). CAOD has been widely used as an assessment tool for occupational dysfunction.

*SRS-18 (Kazuhito et al., 2013).* The SRS-18 was used to measure job related stress response using 18 items in 3 subscales: depression and anxiety (6 items), displeasure and anger (6 items), and lassitude (6 items) with a 4-point response (0 = completely different, 3 = it's correct). High point totals are related to higher degrees of stress.

#### Statistical analysis

*Sample characteristics.* The participants' demographics were summarized using descriptive analyses. The normal distribution of all scores was analyzed using the Kolmogorov–Smirnov test ($p < 0.05$).

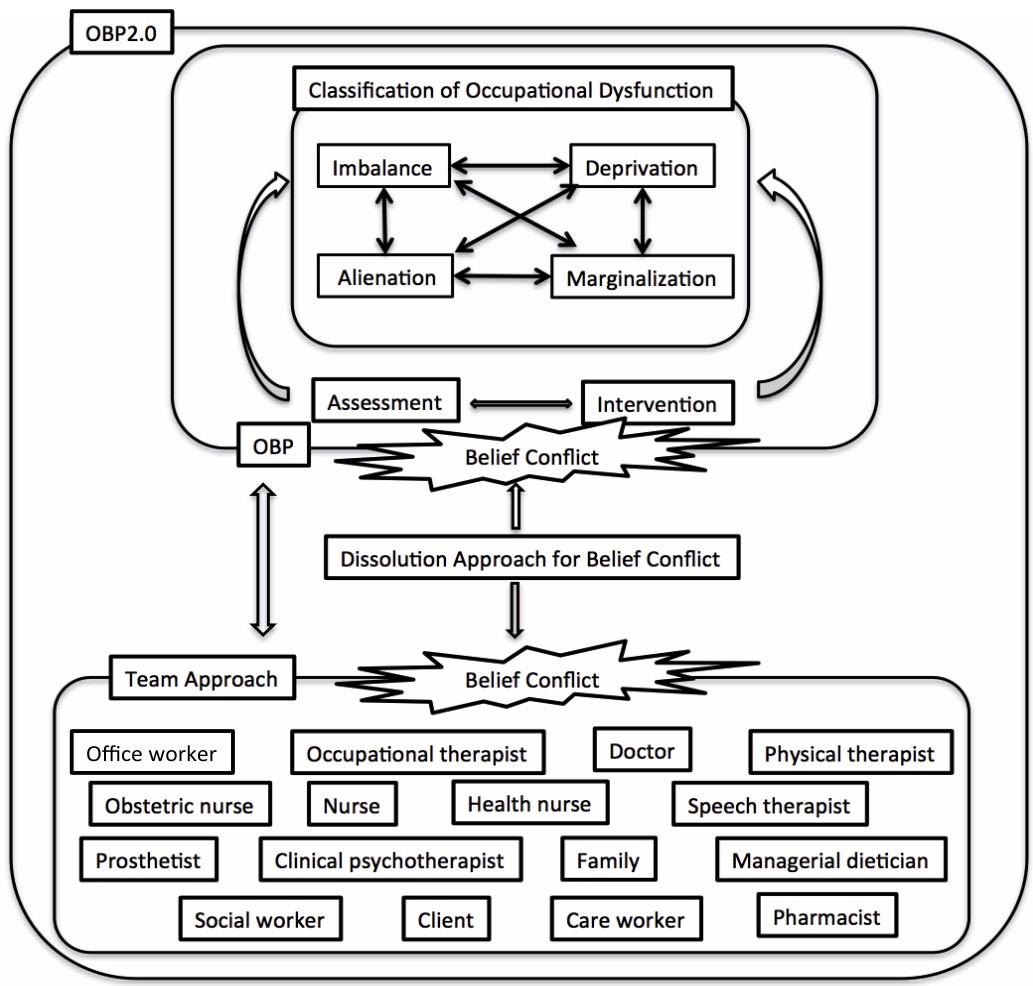

**Figure 2  Occupation based practice 2.0 (OBP2.0) model.** Note. Description of theoretical structure of the OBP2.0. The purpose of this model is to improving of occupational dysfunction and promoting of teamwork. The improving of occupational dysfunction has used assessment and intervention of an occupation based practice (OBP). OBP is occupational therapy technique to increase with health and well-being through a meaningful occupation. In addition, using this model, the promoting of teamwork has used the Dissolution Approach for Belief Conflict (DAB) (*Kyougoku et al., 2015*). DAB is intervention technique for dissolving the dissensus. OBP2.0 is able to use for improve the both occupational dysfunction and belief conflict.

*Structural validity.*  The factor structure of CAOD and SRS-18 was determined by confirmatory factor analysis (CFA) using a robust weighted least squares factoring method (WLSMV) with missing data (*Asparouhov & Muthén, 2010*). The WLSMV is robust to deviations of data from a hypothetical model and is recommended for structural equation modeling of categorical data with non-normality. Therefore, we have selected the estimation for WLSMV.

We utilized three indexes to evaluate the model data fit of CFA. The first and second indexes were the comparative fit index (CFI) and the Tucker-Lewis index (TLI), with critical values above 0.90. The third index was the root mean square error of approximation

(RMSEA). The diagnostic values of RMSEA from 0.08 to 0.10 indicate a modest fit while less than 0.08 indicate a good fit (*Kline, 2011*).

If an unacceptable model fit by CFA was found, we performed an exploratory factor analysis (EFA), utilizing a WLSMV. EFA is able to select an appropriate factor structure through reanalysis even where a poor model fit by CFA was found (*Kline, 2011*). EFA also utilizes CFI, TLI, and RMSEA to estimate the model data fit. We reanalyze factor structure of assessment tool by CFA based on the factor structure supported by the EFA (*Kline, 2011*; *Timothy, 2015*).

*Correlation analysis.* Correlation analysis was assessed using Spearman's correlation coefficient to measure the association between the factor and total score of CAOD, SRS-18, and the personal factors. The covariates were entered into the path analysis using a correlation analysis technique. Personal factor of more than 0.2 of correlation coefficient against all tools was considered as statistically significant covariates.

*Structural model.* The hypothetical model was analyzed using SEM by WLSMV with missing data. The analysis was examining the structural relationship of occupational dysfunction on the stress response in Fig. 1. Personal factors were considered to be covariates that influenced the SRS-18 and CAOD scores. We evaluated the model fit of the hypothesized relationships between latent variables (occupational dysfunction, stress response, and personal factors) and data from SEM. The indirect effects path was a stress response to covariates including occupational dysfunction.

Model fit index used the CFI, TLI, and RMSEA. The critical values of RMSEA from 0.08 to 0.10 indicate a mediocre fit, and below 0.08 indicate a good fit. The critical values for CFI and TLI were 0.90 and above. The significance of standardization coefficient were examining by $p$ value ($p < 0.05$) and 95% confidence interval (95% CI). This model also estimated the indirect effects path, using Mplus.

## Results
### Sample characteristics
Sample characteristics are indicated in Table 1. Participants' average age was 35.8 (SD = 10.2) with a gender distribution of 141 (30.1%) males, 317 (67.7%) females, and 10 (2.1%) others. The Kolmogorov–Smirnov test indicated that all scores had a normal distribution.

### Structural validity of CAOD and SRS-18
Analysis of the CAOD using CFA found that four factors were a good model fit (RMSEA = 0.097, CFI = 0.963, and TLI = 0.954) in Fig. 3. Figure 4 shows the results of the CFA of the SRS-18. The three factors of the SRS-18 were estimated to be a good model fit (RMSEA = 0.089, CFI = 0.951, and TLI = 0.943).

### Correlation analysis
Results are shown in Table 2. Age, gender, job category, and years of work experience had no relation to the CAOD total score and SRS-18. In addition, significant correlation was observed between occupational dysfunction and limited opportunities for refreshment,

**Table 1** Sample characteristics of CAOD and SRS-18 (study 1).

| | | M ± SD |
|---|---|---|
| Age | Total | 35.8 ± 10.2 |
| | Doctors | 48.6 ± 10.2 |
| | Nurses | 36.1 ± 10.8 |
| | Physical therapists | 34.4 ± 8.5 |
| | Occupational therapists | 33.7 ± 8.3 |
| | Others | 34.9 ± 9.3 |
| Years of work experience | Total | 11.8 ± 9.4 |
| | Doctors | 23.3 ± 10.0 |
| | Nurses | 13.5 ± 10.2 |
| | Physical therapists | 10.9 ± 7.6 |
| | Occupational therapists | 9.62 ± 7.7 |
| | Others | 9.68 ± 8.3 |

| | | Total $N$ | % |
|---|---|---|---|
| Gender | Male | 141 | 30.1 |
| | Female | 317 | 67.7 |
| | Others | 10 | 2.1 |
| Job category | Doctor | 21 | 4.5 |
| | Nurse, Health nurse, Midwife | 159 | 34.0 |
| | Physical therapist | 52 | 11.1 |
| | Occupational therapist | 60 | 12.8 |
| | Other healthcare workers | 176 | 37.6 |
| Opportunities for refreshment | Very good | 41 | 8.7 |
| | Good | 265 | 56.6 |
| | Neither good nor bad | 52 | 11.1 |
| | Fair | 56 | 11.9 |
| | Poor | 44 | 9.4 |
| | Unknown | 10 | 2.1 |
| Time spent on leisure activities | Very good | 29 | 6.2 |
| | Good | 224 | 34.6 |
| | Neither good nor bad | 56 | 12.0 |
| | Fair | 114 | 24.4 |
| | Poor | 25 | 3.9 |
| | Unknown | 20 | 4.3 |
| Work relationships | Very good | 22 | 4.7 |
| | Good | 107 | 22.9 |
| | Neither good nor bad | 47 | 10.0 |
| | Fair | 19 | 4.1 |
| | Poor | 1 | 0.2 |
| | Unknown | 272 | 58.1 |

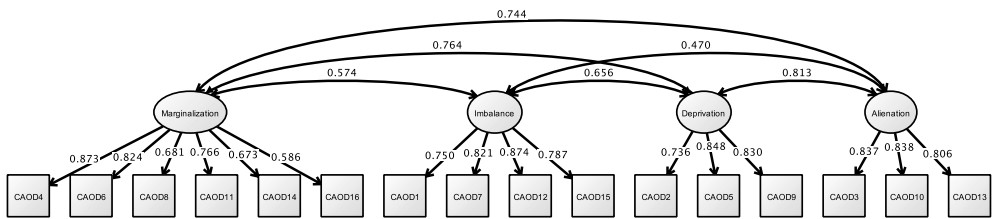

**Figure 3 CFA of CAOD (study 1).** Note. Marginalization, Occupational marginalization; Imbalance, Occupational imbalance; Deprivation, Occupational deprivation; Alienation, Occupational alienation. Previous study showed the occupational imbalance (CAOD1, 7, 12, 15), occupational deprivation (CAOD2, 5, 9), occupational alienation (CAOD3, 10, 13), and occupational marginalization (CAOD4, 6, 8, 11, 14, 16). RMSEA = 0.097, CFI = 0.963, TLI = 0.954.

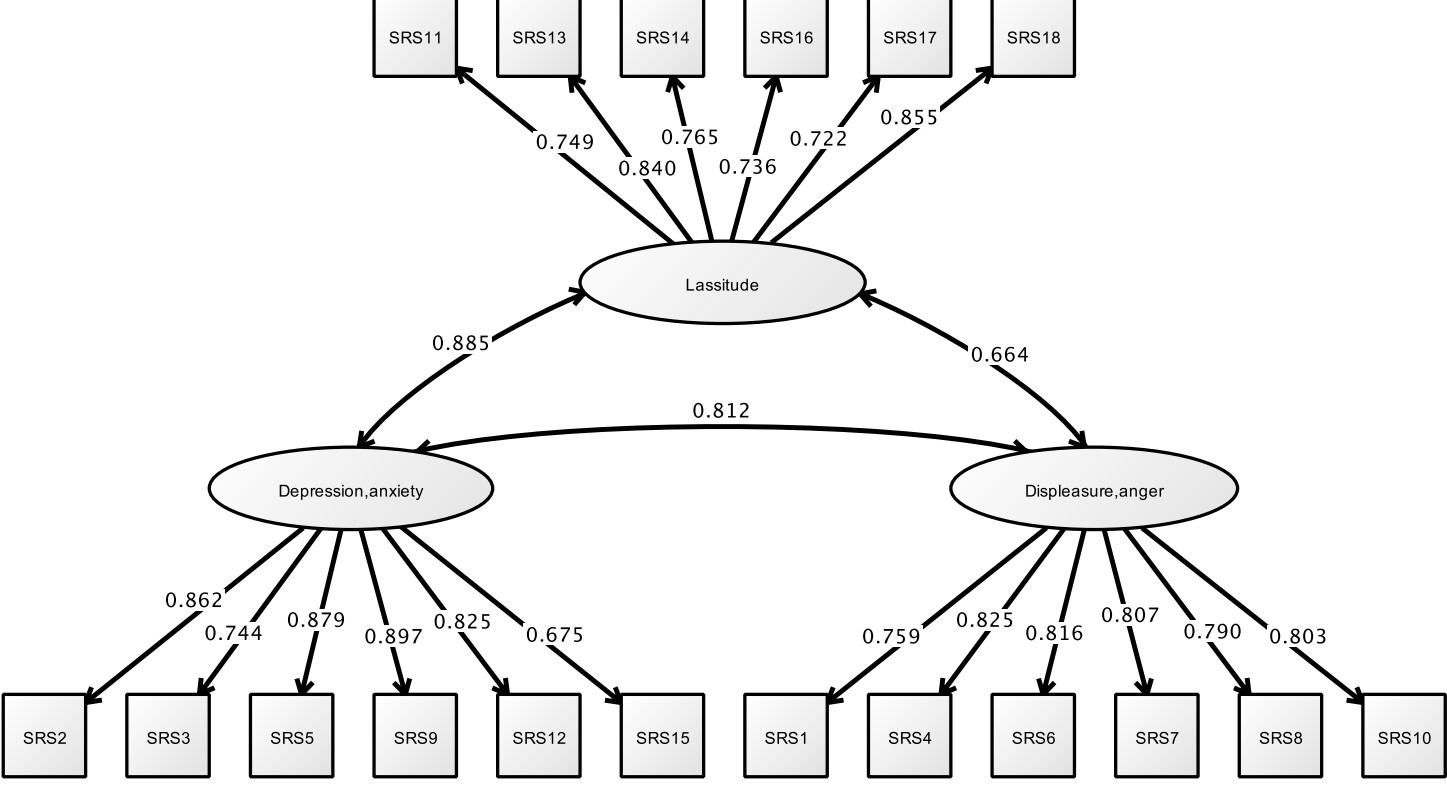

**Figure 4 CFA of SRS-18 (study 1).** Note. Previous study showed the depression and anxiety (SRS2, 3, 5, 9, 12, 15), displeasure and anger (SRS1, 4, 6, 7, 8, 10), and lassitude (SRS11, 13, 14, 16, 17, 18). RMSEA = 0.089, CFI = 0.951, TLI = 0.943.

time spent on leisure activities, and work relationships. These covariates can be understood as problems that individuals feel. Therefore, these variables can be grouped together as latent variables. Table 3 shows the correlation between the factor score of CAOD and SRS-18. The correlation was obtained from all the factors. Therefore, occupational dysfunction and stress response were found to be associated.

### Structural relationship

Figure 5 and Table 4 shows the model fit indicators for structural relationships. Model fit indicators for the structural model demonstrated of model fit (RMSEA = 0.061,

**Table 2 Correlation analysis of personal factors and CAOD total score in study 1, 2, and 3.**

| | | Study 1 | | Study 2 | | Study 3 | |
|---|---|---|---|---|---|---|---|
| | | CAOD | SRS-18 | CAOD | JBS | CAOD | CES-D |
| Age | | −0.09 | −0.122[*] | 0.012 | −0.156[**] | 0.053 | −0.115[**] |
| Gender | | −0.09 | −0.096[*] | −0.065[*] | **0.243**[**] | 0.074 | 0.064 |
| Job category | Nurses | 0.152[*] | −0.134[**] | 0.160[*] | **0.275**[**] | 0.199[**] | 0.079 |
| | Physical therapists | 0.019 | −0.047 | −0.136[**] | −0.070[*] | −0.185[**] | −0.071 |
| | Occupational therapists | −0.092 | −0.029 | −0.110[**] | −0.050 | −0.102[**] | −0.023 |
| | Others | −0.117[*] | −0.002 | −0.009 | −0.021 | −0.017 | . |
| Years of work experience | | −0.049 | −0.063 | 0.044 | −0.127[**] | 0.067 | −0.138[**] |
| Opportunities for refreshment | | **0.530**[**] | **0.309**[**] | **0.485**[**] | **0.224**[**] | **0.463**[**] | **0.313**[**] |
| Time spent on leisure activities | | **0.559**[**] | **0.347**[**] | **0.525**[**] | **0.277**[**] | **0.517**[**] | **0.392**[**] |
| Work relationships | | **0.392**[**] | **0.442**[**] | **0.429**[**] | **0.305**[**] | **0.438**[**] | **0.356**[**] |

Notes.
[*] Significant at 5% level.
[**] Significant at 1% level.
Bold indicates correlation coefficient >0.2.

**Table 3 Correlation analysis between CAOD, SRS-18, JBS, and CES-D.**

| | | Imbalance | Deprivation | Alienation | Marginalization |
|---|---|---|---|---|---|
| St. 1 | Depression and anxiety | 0.342[**] | 0.415[**] | 0.438[**] | 0.529[**] |
| | Displeasure and anger | 0.302[**] | 0.321[**] | 0.357[**] | 0.490[**] |
| | Lassitude | 0.352[**] | 0.476[**] | 0.503[**] | 0.534[**] |
| | Total SRS-18 | 0.382[**] | 0.462[**] | 0.492[**] | 0.583[**] |

| | | Imbalance | Deprivation | Alienation | Marginalization (non-shared) | Marginalization (shared) |
|---|---|---|---|---|---|---|
| St. 2 | Emotional exhaustion | 0.530[**] | 0.417[**] | 0.482[**] | 0.480[**] | 0.222[**] |
| | Depersonalization | 0.343[**] | 0.378[**] | 0.505[**] | 0.524[**] | 0.313[**] |
| | Diminished personal accomplishment | −0.02 | 0.014 | 0.174[**] | 0.021 | −0.029 |
| | Total JBS | 0.306[**] | 0.288[**] | 0.464[**] | 0.376[**] | 0.168[**] |
| St. 3 | Depressed affect | 0.400[**] | 0.392[**] | 0.476[**] | 0.408[**] | 0.251[**] |
| | Somatic symptoms | 0.438[**] | 0.406[**] | 0.481[**] | 0.409[**] | 0.272[**] |
| | Interpersonal difficulties | 0.178[**] | 0.217[**] | 0.264[**] | 0.369[**] | 0.291[**] |
| | Negative affect | 0.177[**] | 0.292[**] | 0.380[**] | 0.302[**] | 0.191[**] |
| | Total CES-D | 0.426[**] | 0.461[**] | 0.581[**] | 0.503[**] | 0.330[**] |

Notes.
St., Study; Imbalance, Occupational imbalance; Deprivation, Occupational deprivation; Alienation, Occupational alienation; Marginalization, Occupational marginalization; Marginalization (non-shared), Non-shared environment marginalization; Marginalization (shared), Shared environment marginalization.
Study 1 is separated 4 factor of CAOD, Study 2 and 3 are separated 5 factor of CAOD.
[**] Significant at 1% level.

CFI = 0.947, and TLI = 0.943). In this model, occupational dysfunction has structural relationship the stress response (standardized direct effect = 0.748, 95% CI [0.500–0.995], $p < 0.001$). Moreover, the covariates (such as opportunities for refreshment, time spent on leisure activities, and work relationships) have structural relationships with occupational
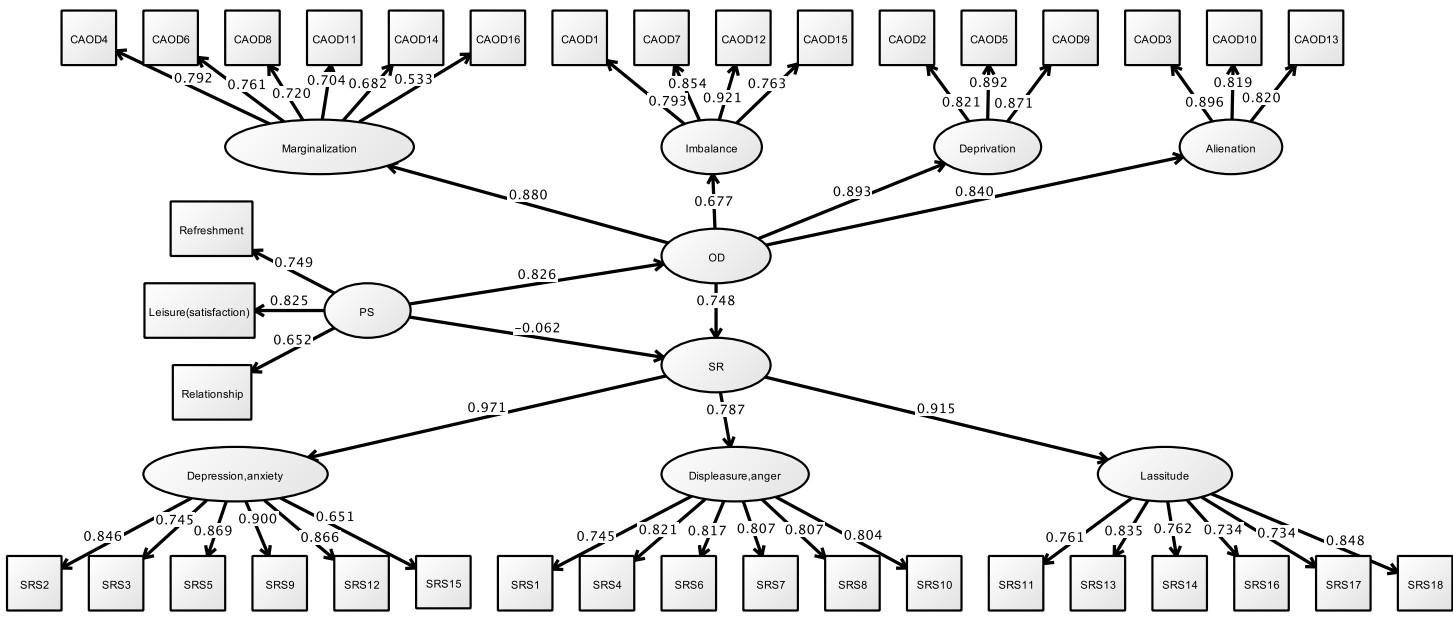

**Figure 5 Structural relationships of SRS-18 on CAOD (study 1).** Note. OD, Occupational dysfunction; PS, Personal factor; SR, Stress response; Refreshment, Opportunities for refreshment; Leisure (satisfaction), Time spent on leisure activities, Relationship, Work relationships. RMSEA = 0.061, CFI = 0.947, TLI = 0.943.

dysfunction (standardized direct effect = 0.826, 95% CI [0.758–0.894], $p < 0.001$). However, these covariates were not found to be related to job stress response (standardized direct effect = −0.062, 95% CI [−0.338–0.215], $p = 0.566$). The indirect effects of SRS of the covariates, including occupational dysfunction, was also estimated = 0.617 (95% CI [0.396–0.838], $p < 0.001$).

# STUDY 2

## Purpose

This study aims to test the hypothesis that burnout syndrome is influenced by occupational dysfunction in healthcare workers (see Fig. 1). Moreover, this hypothesis model examines the effect of personal factors on burnout syndrome and occupational dysfunction.

## Methods

### Participants

There were a total of 1,142 participants (21 doctors, 484 nurses, 205 physical therapists, 180 occupational therapists, and 252 other healthcare workers).

### Measures

*Sample characteristics.* Same as Study 1.

*CAOD (Teraoka & Kyougoku, 2015).* Same as Study 1.

*JBS (Kubo, 2014).* The JBS measures burnout syndrome in three domains: depersonalization (6 items; score range 6–30), emotional exhaustion (5 items; score range 1–25), and

**Table 4  Structural relationships of CAOD and SRS-18.**

|  |  | Estimate | S.E. | Est./S.E. | Two-tailed *P*-value | 95% CI |
|---|---|---|---|---|---|---|
| **Model fit information** |  |  |  |  |  |  |
| RMSEA |  | 0.061 (90% CI [0.057–0.064]) |  |  |  |  |
| CFI |  | 0.947 |  |  |  |  |
| TLI |  | 0.943 |  |  |  |  |
| **Standardized model results** |  |  |  |  |  |  |
| **Stress** | On |  |  |  |  |  |
| Occupational dysfunction |  | 0.748 | 0.096 | 7.771 | 0.000 | 0.500; 0.995 |
| Covariates |  | −0.062 | 0.107 | −0.574 | 0.566 | −0.338; 0.215 |
| **Occupational dysfunction** | On |  |  |  |  |  |
| Covariates |  | 0.826 | 0.026 | 31.284 | 0.000 | 0.758; 0.894 |
| **Occupational dysfunction** | By |  |  |  |  |  |
| Occupational imbalance |  | 0.677 | 0.028 | 23.861 | 0.000 | 0.604; 0.750 |
| Occupational deprivation |  | 0.893 | 0.017 | 53.737 | 0.000 | 0.850; 0.936 |
| Occupational alienation |  | 0.840 | 0.021 | 39.462 | 0.000 | 0.785; 0.894 |
| Occupational marginalization |  | 0.880 | 0.017 | 52.298 | 0.000 | 0.837; 0.923 |
| **Stress response** | By |  |  |  |  |  |
| Depression and anxiety |  | 0.971 | 0.014 | 70.326 | 0.000 | 0.935; 1.006 |
| Displeasure and anger |  | 0.787 | 0.024 | 32.723 | 0.000 | 0.725; 0.849 |
| Lassitude |  | 0.915 | 0.017 | 52.651 | 0.000 | 0.870; 0.960 |
| Covariates | Ind |  |  |  |  |  |
| Stress response |  | 0.617 | 0.086 | 7.195 | 0.000 | 0.396; 0.838 |
| R square |  |  |  |  |  |  |
| Stress response |  | 0.487 | 0.040 | 12.112 | 0.000 | – |

**Notes.**

S.E., Standard error; Est./S.E., Estimator/Standard error; CI, Confidence interval; On, Structural association; By, Constract; Ind, Indirect; R square, R coefficient of determination.

diminished personal accomplishment (6 items; score range 6–30). The JBS comprises 17 items on a 5-point response scale from 1 (disagree) to 5 (agree).

### Statistical analysis

Statistical analysis is the same as Study 1. Measurement tools used: CAOD and JBS.

## Results

### Sample characteristics

Table 5 shows the results of sample characteristics. Participants' average age was $34.5 \pm 10.2$ years with a gender distribution of 476 (41.6%) males, 650 (56.9%) females, and 16 (1.4%) others. The Kolmogorov–Smirnov test showed that all scores were normally distributed.

### Structural validity of CAOD and JBS

Figure 6 shows the results of CFA of CAOD. First, CFA was found to be over the criteria of RMSEA of model fit (RMSEA = 0.102, CFI = 0.951, and TLI = 0.940). Therefore, we used EFA with WLSMV. EFA was showed with five factors that included occupational marginalization of shared environment (2 items), occupational marginalization of

Teraoka
Kyougoku
2015
10.7717/peerj.1389

**Table 5 Sample characteristics of CAOD and JBS (study 2).**

|  |  | M ± SD |  |
|---|---|---|---|
| Age | Total | 34.5 ± 10.2 | |
| | Doctors | 48.6 ± 10.2 | |
| | Nurses | 36.1 ± 10.5 | |
| | Physical therapists | 30.6 ± 7.3 | |
| | Occupational therapists | 29.6 ± 6.6 | |
| | Another | 37.2 ± 11.1 | |
| Years of work experience | Total | 10.5 ± 9.3 | |
| | Doctors | 23.3 ± 10.0 | |
| | Nurses | 12.5 ± 9.7 | |
| | Physical therapists | 7.0 ± 6.4 | |
| | Occupational therapists | 6.4 ± 5.7 | |
| | Another | 11.6 ± 10.5 | |

|  |  | Total N | % |
|---|---|---|---|
| Gender | Male | 476 | 41.6 |
| | Female | 650 | 56.9 |
| | Others | 16 | 1.4 |
| Job category | Doctor | 21 | 1.8 |
| | Nurse, Health nurse, Midwife | 484 | 42.3 |
| | Physical therapist | 205 | 17.9 |
| | Occupational therapist | 180 | 15.7 |
| | Other healthcare workers | 252 | 22.0 |
| Opportunities for refreshment | Very good | 111 | 9.7 |
| | Good | 629 | 55.0 |
| | Neither good nor bad | 143 | 12.5 |
| | Fair | 112 | 9.8 |
| | Poor | 106 | 9.2 |
| | Unknown | 41 | 3.5 |
| Time spent on leisure activities | Very good | 82 | 7.1 |
| | Good | 509 | 44.5 |
| | Neither good nor bad | 33 | 2.8 |
| | Fair | 237 | 20.7 |
| | Poor | 68 | 5.9 |
| | Unknown | 213 | 18.6 |
| Work relationships | Very good | 107 | 9.3 |
| | Good | 463 | 40.5 |
| | Neither good nor bad | 209 | 18.3 |
| | Fair | 48 | 4.2 |
| | Poor | 14 | 1.2 |
| | Unknown | 301 | 26.3 |

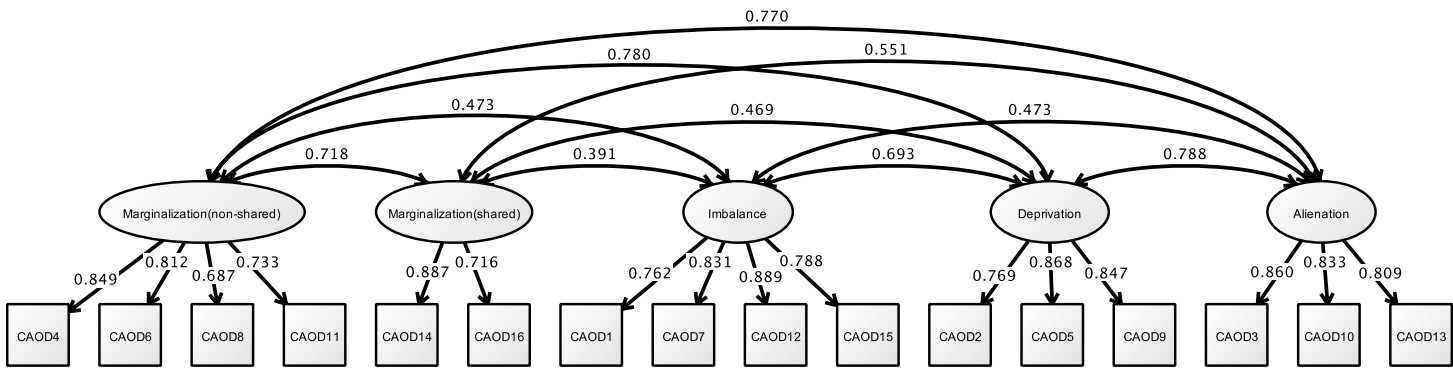

**Figure 6 CFA of CAOD (study 2).** Note. Marginalization (non-shared), non-shared environmental occupational marginalization; Marginalization (shared), shared environmental occupational marginalization. Another latent variables name and factor structure are same as study 1. RMSEA = 0.089, CFI = 0.965, TLI = 0.955.

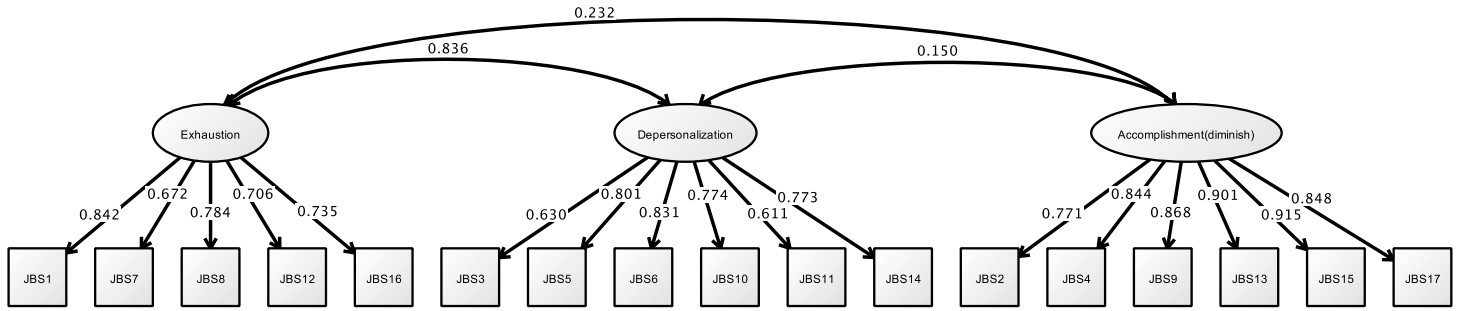

**Figure 7 CFA of JBS (study 2).** Note. Exhaustion, Emotional exhaustion; Accomplishment (diminish), Diminished personal accomplishment. Previous study showed the emotional exhaustion (JBS1, 7, 8, 12, 16), depersonalization (JBS3, 5, 6, 10, 11, 14), and diminished personal accomplishment (JBS2, 4, 9, 13, 15, 17). RMSEA = 0.091, CFI = 0.963, TLI = 0.956.

unshared environment (4 items), occupational imbalance (4 items), occupational alienation (3 items), and occupational deprivation (3 items). Therefore, based on EFA, we performed CFA, and found that the CAOD comprised 16 items with 5 factors. The model fit were RMSEA = 0.089, CFI = 0.965, and TLI = 0.955.

Figure 7 shows the results of the CFA of the JBS. The three factors of the JBS were estimated to be a model fit (RMSEA = 0.091, CFI = 0.963, and TLI = 0.956).

### Correlation analysis

The results are shown in Table 2. Age and years of work experience had no correlation to the CAOD total score or the JBS. Gender and job category of nurses had a weak correlation to JBS. Opportunities for refreshment, time spent on leisure activities, and work relationships fulfilled the criterion correlation. The covariates are the same as the latent variable from Study 1. Table 3 shows the correlation between CAOD and JBS. No or only weak correlations were found between Diminished personal accomplishment and CAOD (includes occupational imbalance, occupational deprivation, occupational marginalization, and total score).

**Table 6  Structural relationships CAOD and JBS (study 2).**

| | | Estimate | S.E. | Est./S.E. | Two-tailed P-value | 95% CI |
|---|---|---|---|---|---|---|
| **Model fit information** | | | | | | |
| RMSEA | | 0.076 (90% CI [0.073–0.078]) | | | | |
| CFI | | 0.919 | | | | |
| TLI | | 0.913 | | | | |
| **Standardized model results** | | | | | | |
| **Burnout** | On | | | | | |
| Occupational dysfunction | | 0.876 | 0.060 | 14.714 | 0.000 | 0.723; 1.029 |
| Covariates | | −0.173 | 0.068 | −2.538 | 0.011 | −0.349; 0.003 |
| **Occupational dysfunction** | On | | | | | |
| Covariates | | 0.796 | 0.018 | 45.214 | 0.000 | 0.750; 0.841 |
| **Occupational dysfunction** | By | | | | | |
| Occupational imbalance | | 0.714 | 0.017 | 41.649 | 0.000 | 0.670; 0.758 |
| Occupational deprivation | | 0.884 | 0.010 | 86.646 | 0.000 | 0.857; 0.910 |
| Occupational alienation | | 0.855 | 0.012 | 70.107 | 0.000 | 0.824; 0.886 |
| Occupational marginalization (non shared) | | 0.888 | 0.011 | 84.104 | 0.000 | 0.861; 0.916 |
| Occupational marginalization (shared) | | 0.615 | 0.024 | 25.447 | 0.000 | 0.552; 0.677 |
| **Burnout syndrome** | By | | | | | |
| Emotional exhaustion | | 0.971 | 0.014 | 67.697 | 0.000 | 0.934; 1.008 |
| Depersonalization | | 0.871 | 0.016 | 55.916 | 0.000 | 0.831; 0.911 |
| Diminished personal accomplishment | | 0.178 | 0.032 | 5.642 | 0.000 | 0.097; 0.260 |
| Covariates | Ind | | | | | |
| Burnout syndrome | | 0.697 | 0.055 | 12.725 | 0.000 | 0.556; 0.838 |
| R square | | | | | | |
| Burnout syndrome | | 0.556 | 0.029 | 19.135 | 0.000 | – |

**Notes.**

S.E., Standard error; Est./S.E., Estimator/Standard error; CI, Confidence interval; On, Structural association; By, Constract; Ind, Indirect; R square, R coefficient of determination.

### Structural relationship

Figure 8 and Table 6 shows that the hypothesized model exhibited fit on the first analysis (RMSEA = 0.076, CFI = 0.919, TLI = 0.913). In this model, occupational dysfunction a structural relationship to burnout syndrome (standardized direct effect = 0.876, 95% CI [0.723–1.029], $p < 0.001$). Moreover, some personal factors as covariates (such as opportunities for refreshment, time spent on leisure activities, and work relationships) are structurally related to occupational dysfunction (standardized direct effect = 0.796, 95% CI [0.750–0.841], $p < 0.001$). However, the covariates were not found to have a highly significant relationship to burnout syndrome (standardized direct effect = −0.173, 95% CI [−0.349–0.003], $p = 0.011$). Indirect effects of JBS to covariates including occupational dysfunction was estimated = 0.697 (95% CI [0.556–0.838], $p < 0.001$).

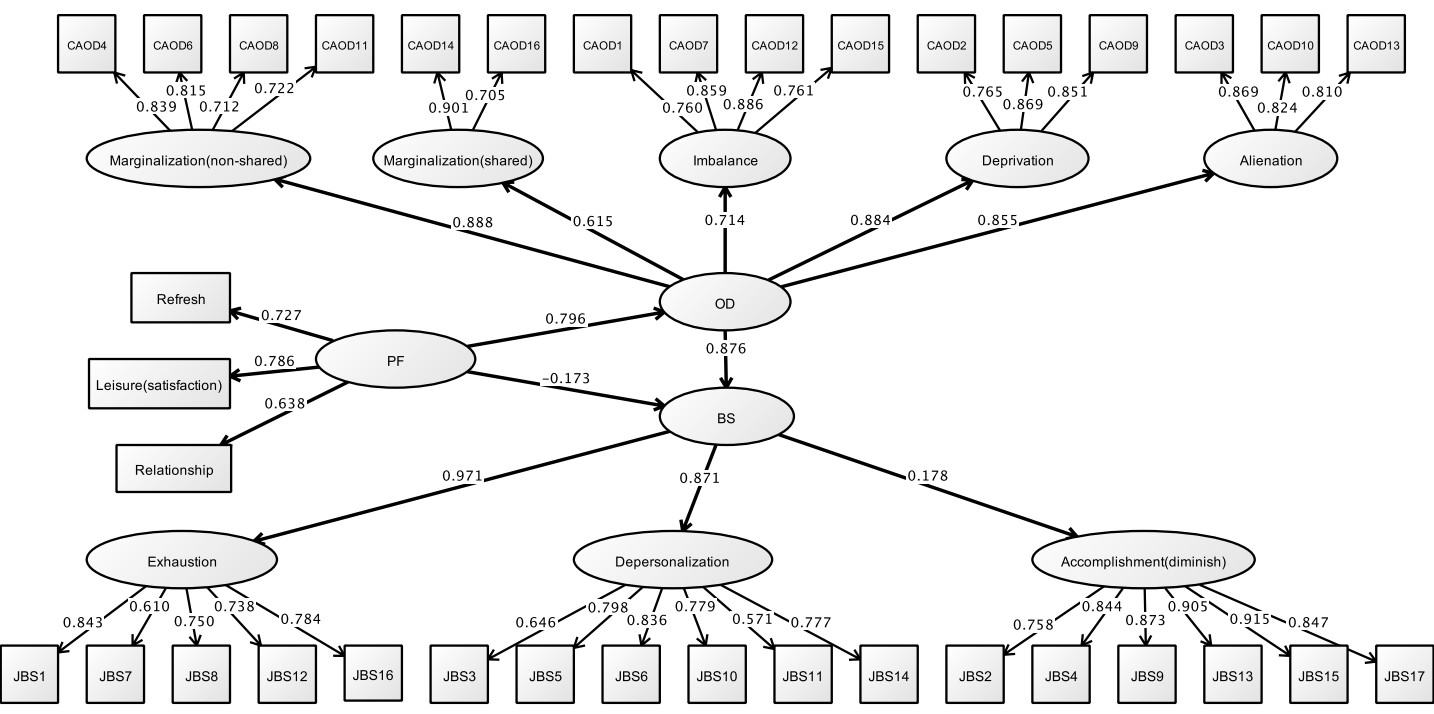

**Figure 8** **Structural relationships of JBS on CAOD (study 2).** Note. OD, Occupational dysfunction; BS, Burnout syndrome. Another latent variables name are same as Study 1. RMSEA = 0.076, CFI = 0.919, TLI = 0.913.

## STUDY 3

### Purpose

This study aims to test the hypothesis that depression is influenced by occupational dysfunction in healthcare workers (see Fig. 1). Moreover, this hypothesis model examines the effect of personal factors on depression and occupational dysfunction.

### Methods
#### *Participants*

In this study, a total of 687 participants were included: 401 nurses (including 12 public health nurses and midwives and 63 assistant nurses), 155 physical therapists, 123 occupational therapists, and 8 other healthcare workers.

#### *Measures*

*Sample characteristics.* Same as Study 1 and 2.

*CAOD (Teraoka & Kyougoku, 2015).* Same as Study 1 and 2.

*CES-D (Shima et al., 1985).* CES-D was measured based on the level of depression experienced within the past week using 20 items on 4 subscales: depressed affect (7 items), negative affect (4 items), interpersonal difficulties (2 items), and somatic symptoms (7 items). Questions were answered using a 4-point response (0 = never, 3 = all the time).

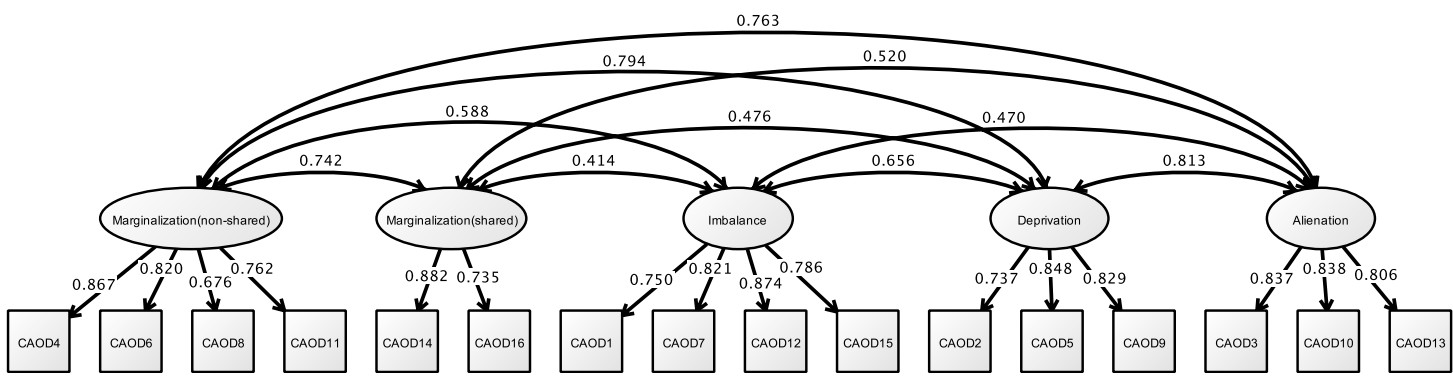

**Figure 9 CFA of CAOD (study 3).** Note. Latent variables name and factor structure are same as study 2. RMSEA = 0.092, CFI = 0.958, TLI = 0.946.

In epidemiologic studies, CES-D has been used worldwide as an assessment tool for depression. Among the negative affect-related items, 4 were originally regarded as related to a positive affect. In the present study, the 4 items were inversely scored to make this point more comprehensible.

### Statistical analysis

Statistical analysis is the same as Study 1. Measurement tools used: CAOD and CES-D.

## Results

### Sample characteristics

Table 7 indicates the results of sample characteristics, including 159 males, 509 females, and 7 unknowns, with an average age of 33.6 ± 10.2 years. The Kolmogorov–Smirnov test showed that all scores had normal distribution.

### Structural validity of CAOD and CES-D

Figure 9 shows the results of the CFA on CAOD. Firstly, the CFA was found to have a poor estimate of RMSEA for model fit (RMSEA = 0.104, CFI = 0.943, and TLI = 0.931). Therefore, we performed EFA, and found that the CAOD comprised 16 items of 5 factors like study 2. The indexes for the EFA model were RMSEA = 0.066, CFI = 0.988, and TLI = 0.972. Based on EFA, CFA of CAOD was determined to be a good estimate of model fit (RMSEA = 0.092, CFI = 0.958, TLI = 0.946).

Figure 10 shows the results of a CFA of CES-D. The CFA model for the latent factors of CES-D exhibited good fit for depressed affect, negative affect, interpersonal difficulties, and somatic symptoms (RMSEA = 0.060, CFI = 0.950, and TLI = 0.942).

### Correlation analysis

The results are shown in Table 2. Age, gender, job category, and years of work experience had no correlation to the CAOD total scores. Opportunities for refreshment, time spent on leisure activities, and work relationships fulfilled a criterion correlation. The covariates are the same as the latent variables from Study 1 and 2. Table 3 shows the correlation between CAOD and CES-D. Correlation was obtained from all the factors; occupational dysfunction and depression were found to be associated.

**Table 7 Sample characteristics of CAOD and CES-D (study 3).**

| | | M ± SD | |
|---|---|---|---|
| Age | Total | 33.6 ± 10.2 | |
| | Nurses | 37.4 ± 11.2 | |
| | Physical therapists | 29.3 ± 6.4 | |
| | Occupational therapists | 27.6 ± 4.1 | |
| Years of work experience | Total | 9.67 ± 9.2 | |
| | Nurses | 12.8 ± 10.3 | |
| | Physical therapists | 5.64 ± 5.3 | |
| | Occupational therapists | 4.84 ± 3.5 | |

| | | Total N | % |
|---|---|---|---|
| Gender | Male | 159 | 23.5 |
| | Female | 509 | 75.4 |
| | Unknown | 7 | 1.0 |
| Job category | Nurse | 326 | 48.2 |
| | Health nurse, midwife | 12 | 1.7 |
| | Assistant nurse | 63 | 9.3 |
| | Physical therapist | 155 | 22.9 |
| | Occupational therapist | 123 | 18.2 |
| | Unknown | 8 | 1.1 |
| Opportunities for refreshment | Very good | 71 | 10.5 |
| | Good | 364 | 54 |
| | Neither good nor bad | 91 | 13.5 |
| | Fair | 56 | 8.3 |
| | Poor | 62 | 9.1 |
| | Unknown | 30 | 4.4 |
| Time spent on leisure activities | Very good | 53 | 7.8 |
| | Good | 285 | 42.2 |
| | Neither good nor bad | 141 | 20.9 |
| | Fair | 123 | 18.2 |
| | Poor | 43 | 6.3 |
| | Unknown | 29 | 4.3 |
| Work relationships | Very good | 85 | 12.6 |
| | Good | 356 | 52.8 |
| | Neither good nor bad | 162 | 24 |
| | Fair | 29 | 4.3 |
| | Poor | 13 | 1.9 |
| | Unknown | 29 | 4.3 |

### Structural relationship

Figure 11 and Table 8 demonstrates the results of the final model. The hypothesized model exhibited model fit (RMSEA = 0.060, CFI = 0.922, TLI = 0.917). In this model, occupational dysfunction has a structural relationship with depression (standardized indirect effect = 0.695, $p < 0.001$, 95% CI [0.521–0.869]). Moreover, personal factors are covariates (such as opportunities for refreshment, time spent on leisure activities,

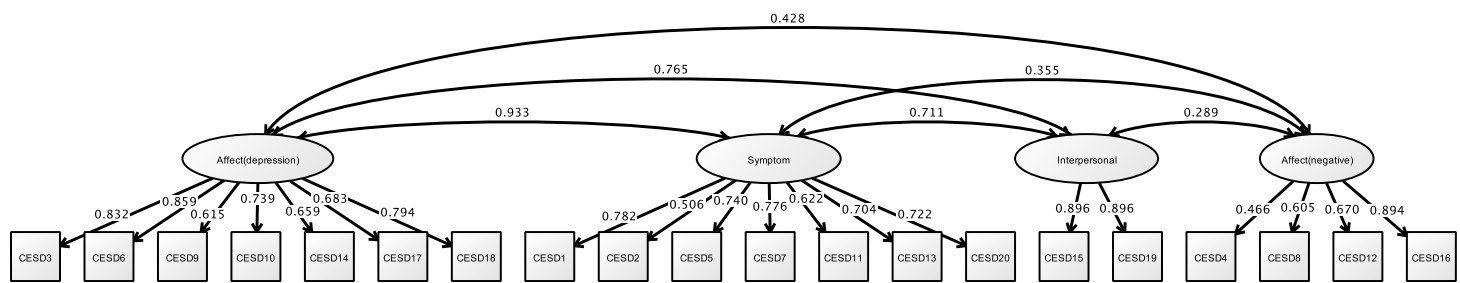

**Figure 10 CFA of CESD (study 3).** Note. Affect (depression), Depressed affect; Symptom, Somatic symptoms; Interpersonal, Interpersonal difficulties; Affect 500 (negative), Negative affect. Previous study showed the depressed affect (CESD3, 6, 9, 10, 14, 17, 18), somatic symptoms (CESD1, 2, 5, 7, 11, 13, 20), interpersonal difficulties (CESD15, 19), and negative affect (CESD4, 8, 12, 16). RMSEA = 0.060, CFI = 0.950, TLI = 0.942.

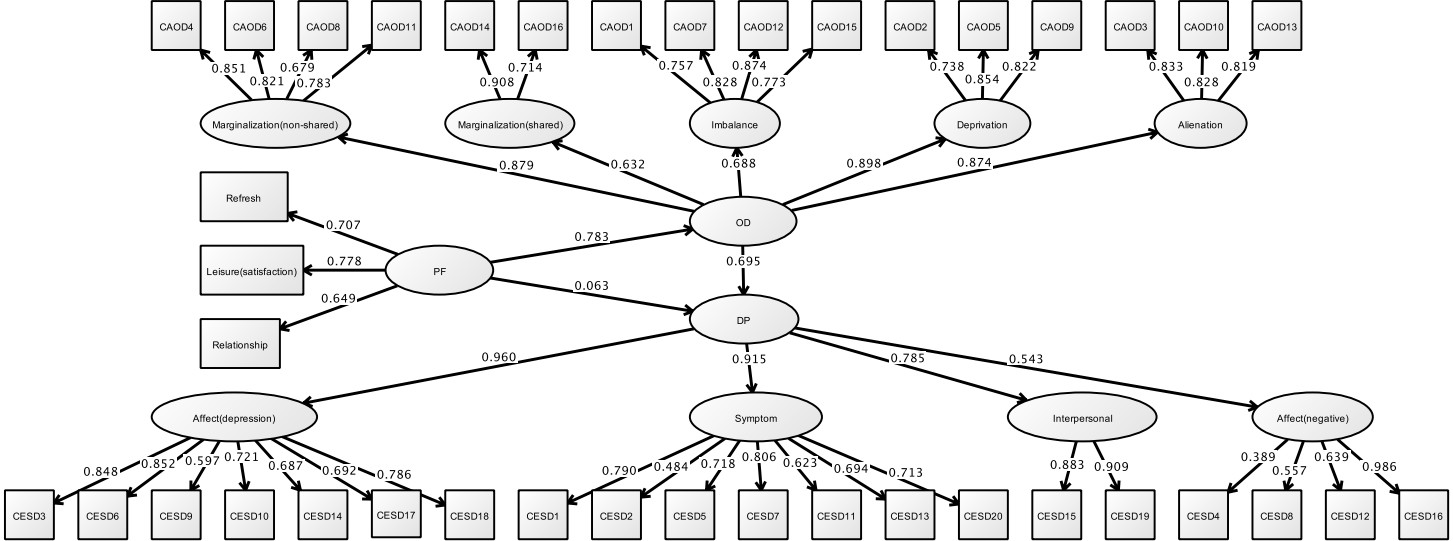

**Figure 11 Structural relationships of CES-D on CAOD (study 3).** Note. DP, Depression. Another latent variables name are same as Study 1 and 2. RMSEA = 0.060, CFI = 0.922, TLI = 0.917.

and work relationships) and have a structural relationship with occupational dysfunction (standardized direct effect = 0.796, 95% CI [0.750–0.841], $p < 0.001$). Furthermore, covariates were not related to burnout syndrome (standardized direct effect = 0.063, 95% CI [−0.133–0.259], $p = 0.407$). Indirect effects of CES-D to covariates including occupational dysfunction were also estimated = 0.544 (95% CI [0.398–0.690], $p < 0.001$).

## DISCUSSION

This study aimed to identify, using hypothetical model of a structural relationship, if psychological problems are affected by occupational dysfunction in healthcare workers (Fig. 1). Our biggest finding was confirming the structural relationship, indicating that hypothesis model was valid across three studies.

Our three studies showed that occupational dysfunction has a significant role on psychological problems; it includes stress, burnout, and depression. We found that CAOD factor scores were significantly and positively correlated with SRS-18, JBS, and CES-D

**Table 8** Structural relationships CAOD and CES-D (study 3).

| | | Estimate | S.E. | Est./S.E. | Two-tailed P-value | 95% CI |
|---|---|---|---|---|---|---|
| **Model fit information** | | | | | | |
| RMSEA | | 0.060 (90% CI [0.057–0.063]) | | | | |
| CFI | | 0.922 | | | | |
| TLI | | 0.917 | | | | |
| **Standardized model results** | | | | | | |
| **Depression** | On | | | | | |
| Occupational dysfunction | | 0.695 | 0.067 | 10.301 | 0.000 | 0.521; 0.869 |
| Covariates | | 0.063 | 0.076 | 0.829 | 0.407 | −0.133; 0.259 |
| **Occupational dysfunction** | On | | | | | |
| Covariates | | 0.783 | 0.023 | 33.791 | 0.000 | 0.723; 0.842 |
| **Occupational dysfunction** | By | | | | | |
| Occupational imbalance | | 0.688 | 0.025 | 27.477 | 0.000 | 0.624; 0.753 |
| Occupational deprivation | | 0.898 | 0.015 | 57.989 | 0.000 | 0.858; 0.938 |
| Occupational alienation | | 0.874 | 0.015 | 56.918 | 0.000 | 0.835; 0.914 |
| Occupational marginalization (non shared) | | 0.879 | 0.014 | 61.800 | 0.000 | 0.843; 0.916 |
| Occupational marginalization (shared) | | 0.632 | 0.032 | 19.721 | 0.000 | 0.549; 0.714 |
| **Depression** | By | | | | | |
| Depressed affect | | 0.960 | 0.015 | 65.960 | 0.000 | 0.922; 0.997 |
| Somatic symptoms | | 0.915 | 0.017 | 54.215 | 0.000 | 0.872; 0.959 |
| Interpersonal difficulties | | 0.785 | 0.032 | 24.395 | 0.000 | 0.702; 0.868 |
| Negative affect | | 0.543 | 0.041 | 13.098 | 0.000 | 0.436; 0.649 |
| Covariates | Ind | | | | | |
| Depression | | 0.544 | 0.057 | 9.578 | 0.000 | 0.398; 0.690 |
| R square | | | | | | |
| Depression | | 0.556 | 0.031 | 17.714 | 0.000 | — |

**Notes.**

S.E., Standard error; Est./S.E., Estimator/Standard error; CI, Confidence interval; On, Structural association; By, Constract; Ind, Indirect; R square, R coefficient of determination.

total scores (Table 3). Moreover, even after making the necessary amendments to the covariance (limited opportunities for refreshment, time spent on leisure activities, and work relationships), psychological problems in healthcare worker were explained by occupational dysfunction (Figs. 5, 8 and 11, Tables 4, 6 and 8). This finding is significant because a majority of research studies on occupational dysfunction have focused on prevalence rate based on epidemiological observational studies (*Akiyama & Kyougoku, 2010*; *Miyake et al., 2014*).

In the cross sectional design, it is difficult to posit cause/effect; however, our results suggest that occupational dysfunction is an important contributing factor in the development of psychological problems. Occupation is the center of the human experience in everyday life; it includes things people need to do, want to do, and are expected to do (*Wilcock & Hocking, 2015*). Occupational dysfunction is a negative aspect of the human lifestyle (*Teraoka & Kyougoku, 2015*). A healthy lifestyle is essential to reducing psychological problems (*Ishii, Kyougoku & Nagao, 2010*). In other words, people

have the ability to promote or reduce psychological problems caused by occupational dysfunctions (*Wilcock & Hocking, 2015*). In this respect, the present findings suggest, by statistical evidence, that occupational dysfunction and psychological problems are significantly structurally related.

This study indicated that all measurement tools were significantly and positively correlated with opportunities for refreshment, time spent on leisure activities, and work relationships in personal factors (Table 2). Meanwhile, the path analysis of three studies indicated that CAOD was only significantly and positively related with these personal factors (Figs. 5, 8 and 11, Tables 4, 6 and 8). In addition, no or only weak correlations were found between all measurement tools and other personal factors (age, gender, job category, and years of work experience) in this study (Table 2). Therefore, based on the conclusions in this study, we believe that behind the occupational dysfunction is a problem of human relations and lack of balanced lifestyles.

The CFA approach displayed good fitness levels. We understood the objective phenomenon using the factor structure of all measurement tools. However, the factor structure of occupational marginalization of the CAOD differed across the three studies. Study 1 and the previous study (*Teraoka & Kyougoku, 2015*) used the same factor structure of occupational marginalization. Studies 2 and 3, however, used different factor structures; one used environmental occupational marginalization and the other did not. However, the environmental occupational marginalization factor stems from the concept of occupational marginalization. Therefore, the CAOD factor structure of study 1 is not irrelevant to that of Studies 2 and 3. We think that the results of CAOD of the three studies can understand as the framework in a similar occupational marginalization.

## Clinical usefulness

CAOD is an assessment tool that was developed as a theoretical background to OBP2.0 (*Teraoka & Kyougoku, 2014*; *Teraoka & Kyougoku, 2015*; *Kyougoku et al., 2015*). It can be used for both people with disabilities as well as healthy people. We believe that intervention with healthcare workers with occupational dysfunction reveals a structural relationship between the occupational dysfunction and psychological problems. To address this, healthcare workers could be routinely asked to answer the CAOD to identify their existing classification of occupational dysfunction. Subsequently, an occupational therapist or occupational health physician could meet with each healthcare worker to review their responses and gain a clear understanding of their occupational dysfunctions. Some of the treatments for occupational dysfunction include occupational therapy, psychological therapy, and cognitive behavioral therapy. Effective application of these approaches could help healthcare workers with occupational dysfunction.

## Limitation

Our study has several limitations. First, this study was cross-sectional design. This design was appropriate because previous studies did not investigate the relationships between the occupational dysfunction and psychological problems; these include stress response, burnout syndrome, and depression. For the future, longitudinal studies are needed to

determine the causal relationship of the existence of occupational dysfunction related psychological problems. Second, all participants were recruited from healthcare workers in Japan. This may limit our ability to generalize these findings to other populations. Third, our study used self-reported assessments; it included CAOD, SRS-18, JBS and CES-D. These instruments have high validity, however, we also needed to use observation assessment for accurate diagnostics.

### Funding
The authors received no funding for this work.

### Competing Interests
The authors declare there are no competing interests.

### Author Contributions
- Mutsumi Teraoka conceived and designed the experiments, performed the experiments, analyzed the data, contributed reagents/materials/analysis tools, wrote the paper, prepared figures and/or tables, reviewed drafts of the paper.
- Makoto Kyougoku conceived and designed the experiments, performed the experiments, analyzed the data, contributed reagents/materials/analysis tools, reviewed drafts of the paper.

### Human Ethics
The following information was supplied relating to ethical approvals (i.e., approving body and any reference numbers):

The Ethics Committee of Kibi International University and the research ethics committee of partnership hospitals approved all research protocol and informed consent procedures (No. 13–30). Written informed consent was obtained from all the participants. We provided participants with a letter explaining the outline and purpose of the study. Participants had the right to drop out of the research project at any time without any reason. We regarded the return of the survey as consent for participation in this research. Surveys were returned in anonymous, sealed envelopes.

### Supplemental Information
Supplemental information for this article can be found online at http://dx.doi.org/10.7717/peerj.1389#supplemental-information.

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
