# Peer review of "Analysis of structural relationship among the occupational dysfunction on the psychological problem in healthcare workers: a study using structural equation modeling"

_PeerJ, doi:10.7717/peerj.1389_

## Round 0.1 · original submission · Major Revisions

Your manuscript entitled "Influence of occupational dysfunction on the psychological problem in healthcare workers: A study using structural equation modeling" has now been seen by the reviewers, whose comments are appended below. You will see that, while they find your work of interest, they have raised points that need to be addressed by a major revision.

Reviewer 1 ·

Basic reporting

1. The occupation based practice 2.0 (OCP2.0) framework references (ref 10, 11, 12) were Japanese (self-referencing by authors), or book texts not journal articles. For this reason, it is difficult to validate it's utility. I advise referencing other journal articles which have evaluated the OCP2.0 validity - ideally these references should be not self-referencing by the authors, and would be available in English or with English translation.

2. To expand on point 1. I advise using figure 1 to explain the OCP2.0 model in greater detail in conjunction with greater explanation in the text. This should be with greater explanation in the text, and with more references as above.

3. For figure 2, suggest elaboration within the figure itself what factors 1, 2, 3, and 4 are with specific examples. Similarly, for figures 3 through 11, I suggest explaining what each 'item' is within these figures - either within the figures themselves, or within the text. I suggest for all figures to avoid the use of underscores between words/terms.

4. The study is a self-contained body of work. However, it is worth considering condensing the paper by combining methods of studies 1, 2, and 3 together, with separate explanations where required. This would avoid needless repetition within 'sample characteristics', 'CAOD', 'statistical analysis', 'correlation analysis', 'testing causal relationships'.

5. Line 209. Term: Mplus. This term is not referenced to earlier and has not definition. Please expand.

Experimental design

1. The experimental design should be complimented. However, it's nature is cross-sectional, and therefore it is impossible to draw conclusions suggesting causal relationship as no temporal analysis is possible See below within 'validity of findings' for greater explanation on this matter.

2. The purpose of each of the studies queries whether there is a causal relationship between occupational dysfunction and stress response, burnout, and depression. However, as above, the cross-sectional design does not allow for this. Instead, only correlation can be determined.

Validity of the findings

1. The conclusions are clear in that a positive correlation is clear between occupational dysfunction with stress response, burnout, and depression. The authors should be more clear in their wordings to avoid stating matters of speculation as fact.

2. For instance in study one 'discussion', lines 237 and 238. It is impossible to state a causal relationship between occupational dysfunction and the stress response. The study by design cannot determine if occupational dysfunction leads to stress response, or indeed if the stress response leads to occupational dysfunction. Similar issues are identified within the discussion/results of study 2 and 3.

Additional comments

Thank you for your contribution to the literature with this important piece of work. Key issues to improve in as above - namely better explanations and referencing for the OCP2.0 model, and to clarify between speculations and findings in regards to determining correlation (Verses causation) with occupational dysfunction and stress response, burnout, and depression.

·

Basic reporting

In my view, there were three problems with the reporting in the Introduction, and I list them here in descending order of importance:

1. I had trouble differentiating several concepts in the Introduction: occupational dysfunction, job stress, psychological problems. These terms eventually got defined sufficiently later in the manuscript, but I think they should be clearly delineated early on to avoid confusion.

2. The authors state that "...there has been no previous study examining the impact of occupational dysfunction on psychological problems, including stress response, burnout syndrome, and depression" (lines 116-118). That must be incorrect. I assume a considerable volume of research has addressed these issues. Thus, in its current form, the article fails to demonstrate how the work fits into the broader field of knowledge.

3. The Introduction is repetitive. I think it could be shortened considerably. For instance, the paragraph starting on line 116 seems to be information already reviewed in the Introduction.

Additionally, I don't think a Discussion section is needed for each separate study; one "General Discussion" should suffice. The authors copied and pasted large segments of the Study 1 Discussion for use in the Study 2 and 3 Discussions.

Experimental design

In this section, I will describe some of the problems with the authors' implementation of a latent variable modeling approach and their reporting of the results. These problems lead me to advise strongly that the article not be published in its present form. In general, I suggest that the authors consult a CFA textbook while revising the manuscript. I recommend Tim Brown's text:

Brown, T. A. (2015). Confirmatory factor analysis for applied research. Guilford Publications.

1. The authors did not explain why they formed a latent variable from their covariates. What is the rationale behind a latent trait underlying "opportunities for refreshment, time spent on leisure activities, and work relationships"?

2. The authors' practice of refining CFA models by conducting EFA is unusual. Typically, if EFA is conducted, it precedes CFA, and it signals that no strong theory exists regarding the number or nature of the factors. That does not seem to be the case here. Instead of EFA, modification indices are usually inspected to refine/rescue ill-fitting CFA models.

More importantly, the authors never describe (a) how the EFAs were conducted or (b) how the EFA results were used to inform CFA respecifications. For instance, how did the authors decide on the optimal number of factors in the EFAs? Parallel analysis? Scree plot? And how did they judge which indicators should be respecified as loading on factors other than the ones on which they were theorized to load?

3. What is the rationale for the WLSMV estimator? It's normally used with categorical or ordered indicators, yet nearly all indicators in the present study would be considered continuous variables.

4. The authors make some truly puzzling interpretations of model fit indices. For instance, on line 224, they claim that a 5-factor CFA of the CAOD in Study 1 (RMSEA = .085, CFI = 0.973, TLI = 0.965) fits better than a 4-factor CFA (RMSEA = .097, CFI = 0.963, TLI = 0.954). That's obviously incorrect. Also, they claim that a 5-factor model is superior to a 4-factor model in Study 2, but then in the Discussion state that "The consequence of CFA showed that the factor structure of these assessments was appropriate" (line 361-362), which I interpreted as saying that "the hypothesized (4-factor) structure of the CAOD was supported," which is obviously not true according to their reading of the fit indices.

5. They authors showed a poor understanding of conventions for acceptable model fit: "The hypothesized model exhibited excellent fit (RMSEA = .060, CFI = 0.922, TLI = 0.917" (line 453), despite citing the guidelines--which they did not follow--three separate times (once in each "Statistical Analysis" subsection of the Methods).

6. The authors often used a "greater than" sign when a "less than" sign was appropriate (e.g., p > .001 when they meant p < .001).

Validity of the findings

In the Limitations section, the authors acknowledge that the study was cross-sectional, yet throughout the manuscript (including the Abstract), they make claims about causality. The old adage is "correlation does not equal causation." All language about directionality and causal relations should be removed. Accordingly, the "Clinical Implications" section, which relies on causal claims, should be rewritten.

Additional comments

The manuscript has several positive features, including large sample sizes and psychometrically sound measures. However, the misuse of latent variable modeling techniques limits confidence in the findings. I recommend either (a) consulting a textbook on CFA or SEM before revising the manuscript, or (b) leaving the latent variable framework entirely and simply reporting the correlations between the observed measures of occupational dysfunction, stress response, burnout, and depression.

I hope my comments do not come across as too harsh, and I hope the suggestions are useful in your revision. I see no reason why results from these datasets cannot be published in PeerJ if you were to considerably improve your analyses and reporting.

---

## Round 0.2 · accepted · Accept

Dear authors:

I have read carefully all the responses to the previous comments, and I think the manuscript has high standards and is well written. It can be published in PeerJ.